# From Fruit Waste to Medical Insight: The Comprehensive Role of Watermelon Rind Extract on Renal Adenocarcinoma Cellular and Transcriptomic Dynamics

**DOI:** 10.3390/ijms242115615

**Published:** 2023-10-26

**Authors:** Chinreddy Subramanaym Reddy, Purushothaman Natarajan, Padma Nimmakayala, Gerald R. Hankins, Umesh K. Reddy

**Affiliations:** Department of Biology, Gus R. Douglass Institute, West Virginia State University, Institute, WV 25112, USA; subramanyam.chinreddy@wvstateu.edu (C.S.R.); pnatarajan@wvstateu.edu (P.N.); padma@wvstateu.edu (P.N.)

**Keywords:** human renal adenocarcinoma cells 769-P, cell proliferation, watermelon rind extract, transcriptome, apoptosis

## Abstract

Cancer researchers are fascinated by the chemistry of diverse natural products that show exciting potential as anticancer agents. In this study, we aimed to investigate the anticancer properties of watermelon rind extract (WRE) by examining its effects on cell proliferation, apoptosis, senescence, and global gene expression in human renal cell adenocarcinoma cells (HRAC-769-P) in vitro. Our metabolome data analysis of WRE exhibited untargeted phyto-constituents and targeted citrulline (22.29 µg/mg). HRAC-769-P cells were cultured in RPMI-1640 media and treated with 22.4, 44.8, 67.2, 88.6, 112, 134.4, and 156.8 mg·mL^−1^ for 24, 48, and 72 h. At 24 h after treatment, (88.6 mg·mL^−1^ of WRE) cell proliferation significantly reduced, more than 34% compared with the control. Cell viability decreased 48 and 72 h after treatment to 45% and 37%, respectively. We also examined poly caspase, SA-beta-galactosidase (SA-beta-gal), and wound healing activities using WRE. All treatments induced an early poly caspase response and a significant reduction in cell migration. Further, we analyzed the transcript profile of the cells grown at 44.8 mg·mL^−1^ of WRE after 6 h using RNA sequencing (RNAseq) analysis. We identified 186 differentially expressed genes (DEGs), including 149 upregulated genes and 37 downregulated genes, in cells treated with WRE compared with the control. The differentially expressed genes were associated with NF-Kappa B signaling and TNF pathways. Crucial apoptosis-related genes such as BMF, NPTX1, NFKBIA, NFKBIE, and NFKBID might induce intrinsic and extrinsic apoptosis. Another possible mechanism is a high quantity of citrulline may lead to induction of apoptosis by the production of increased nitric oxide. Hence, our study suggests the potential anticancer properties of WRE and provides insights into its effects on cellular processes and gene expression in HRAC-769-P cells.

## 1. Introduction

Cancer is a pressing global public health concern and ranks notably as the second most common cause of death in the United States [1,2]. The year 2023 has presented alarming statistics, as the US National Center for Health Statistics, in collaboration with the Center for Disease Control and Prevention, projected that a staggering 81,800 new cases of kidney cancer have surfaced, culminating in approximately 14,890 deaths [3]. An unsettling aspect of cancer treatment lies in the severe adverse effects of numerous anticancer drugs. These effects primarily stem from the drugs’ inability to discern between regular and cancerous cells [4]. Nevertheless, there is a silver lining. By adopting simple lifestyle modifications such as dietary changes, maintaining optimal body weight, and regular physical exercise, a significant portion of cancers (ranging from 30% to 40%) can be potentially thwarted [5]. An encouraging revelation is that an increased consumption of vegetables and fruits might hold the key to averting approximately 20% of annual cancer-related mortalities. The rationale lies in their intrinsic safety, minimal toxicity, potent antioxidant properties, and widespread acceptance as dietary supplements. Thus, in the current era of cancer research, the potential benefits of fruits and vegetables have become a focal point of scientific inquiry [1]. This has spurred medical professionals worldwide to intensively explore novel anticancer compounds sourced naturally and to delve into complementary herbal therapies [6,7].

One such fruit garnering attention is the watermelon. Hailing from the Cucurbitaceae family, watermelon is a prominent horticultural crop cherished for its succulent fruits [8]. While it is a natural diuretic and finds its way into various culinary delights, its rind (WR) is often discarded. Extracting bioactive compounds from this waste could revolutionize the agricultural food chain and present sustainable solutions [7]. Previous research has illuminated the myriad of benefits WR holds, attributing its antioxidant, free radical scavenging, and anti-microbial capabilities to the array of phenolic compounds it houses, such as quercetin, myricetin, and more [9,10,11]. Intriguingly, citrulline, a non-essential amino acid often likened to Viagra, is found in greater abundance in WR than in the pulp, further underscoring its potential health benefits [7,12,13].

### Objectives

Given the potential benefits and therapeutic properties associated with watermelon, especially its rind, our study seeks to analyze watermelon rind waste using LC-MS to discern and comprehensively document the composition of watermelon rind extract (WRE) to investigate the biological effects of WRE with a specific focus on its implications for cell proliferation, apoptosis, cell migration, and the global transcriptomic profile in HRAC-769-P cells, and to establish a comparative understanding between existing research findings on WRE’s inhibitory effects on human cancer cell proliferation and its effects on HRAC-769-P cells. Through our research, we endeavor to bridge existing knowledge gaps, shed light on the implications of WRE on HRAC-769-P cells, and provide a comprehensive understanding of the therapeutic potential of watermelon rind.

## 2. Results

### 2.1. Metabolite Analysis of Phytoconstituents of Watermelon Rind Extraction

WR metabolome analysis identified targeted citrulline (22.29 µg/mg) and other untargeted metabolites in our study, including amino acid derivatives, organic acid derivatives, sugar derivatives, and hydroxycinnamic acid derivatives presented in Table 1.

### 2.2. HRAC-769-P Cell Proliferation Was Affected by Concentration and Duration of Treatment

Various concentrations of WREs’ effects on HRAC-769-P cell proliferation showed dose-dependent cell viability (Figure 1). WRE chemical compounds could affect cell proliferation via L-Citrullin and other chemical constituents. In previous studies, it was observed that the anticancer effects of WRE in vitro were dependent on both the dose and time of exposure [7]. The longer duration of exposure was found to increase the potency of WRE and enhance its stability under specific experimental conditions. The percentage of HRAC-769-P cells decreased as the concentration of WRE increased 24, 48, and 72 h after treatment. (Figure 1). At 24 h after treatment, the proportion decreased to 15%, 33%, 40%, 47%, 53%, 59%, and 66%, respectively, compared with controls. Moreover, further cell growth was inhibited significantly at 48 and 72 h post treatment, to 28%, 54%, 69%, 83%, 88%, 93%, and 95% and 39%, 62%, 76%, 91%, 93%, 95%, and 96%, respectively, compared with controls at various concentrations (Figure 1). 

### 2.3. Apoptosis versus Senescence

Poly caspase activity was noticed from 0.5 h compared with the control in HRAC-769-P cells (Figure 2). This trend continued up to 2 to 4 h. Compared with the control, all concentrations of WRE (44.8 and 88.6 mg) exhibited caspase activity. Compared with the (ethanol-0.96%) control, peak poly caspase activity occurred with WRE (88 mg·mL^−1^) at 2 h, followed by 1 h, respectively. The early induction of caspase activity was noticed, and it began to decrease at 4 h. SA-beta-gal activity with WRE 88.6 mg·mL^−1^ treatment exhibited similarity to control(s) activity at 0.5 and 1 h, followed by a decline (Figure 3). Interestingly, the activity was significantly lower in other treatments between 0.5 and 2 h. Overall, SA-beta-gal activity began to decrease from 2 h in all the treatments.

### 2.4. Assessment of Cell Migration Inhibition and Metastasis Using Wound Healing Assay

The effects of WRE on the progression and migration of HRAC-769-P cells were evaluated for 48 h. The wound in the control group was healed 48 h after scratching a monolayer of cells. When cells were treated with WRE IC50, the wound healing of the scratched area was significantly delayed compared with untreated and WRE-treated cells (Figure 4A). The WRE significantly decreased the wound closure by 93.14% (*p* < 0.00001) in cells compared with that of the control group (Figure 4B). An in vitro cell migration analysis reveals metastatic potential. Cancer metastasis is the foremost reason for cancer death globally [7,14]. In our study, the WRE significantly inhibited cell migration in HRAC-769-P cells. Therefore, WRE components may include potent anticancer agents in decreasing cancer metastasis.

### 2.5. RNAseq Assessment of Differentially Expressed Genes between WRE-Treated and Control Cells

A total of 26,784,649, 27,470,824, and 25,588,547 raw reads were generated from the control condition of HRAC-769-P cells, whereas 32,424,898, 27,827,568, and 31,744,771 reads were generated with WRE treatment (44.8 mg·mL^−1^), respectively. The raw reads were subjected to stringent quality filtering by using a trimmomatic tool, which resulted in 24,702,469, 25,318,621, 23,597,931, 29,939,262, 25,713,431, and 29,270,759 high-quality reads for control and treatment (44.8 mg), respectively. The Q30 percentage of reads in each library was >96%. The reads from the one controlled and two treatment conditions were aligned to the human reference genome (GRCh38.p13) using the STAR universal RNA sequence alignment tool with default parameters. Then, 96.4, 96.3, 96.4, 96.5, 96.4, and 96.3 percent quality-filtered reads were mapped to the reference genomes for WRC and WRT44, respectively; ~3.6% of the reads remained unmapped (Table 2).

The volcano plot in (Figure 5) shows each treatment’s total up- and downregulated genes based on −log10 (*p*-value) and log2 fold change. It includes 149 upregulated and 37 downregulated genes. The top 10 up- and downregulated DEGs shared among the two treatments are presented in Appendix A.

In total, 33 (up) and 24 (down) statistically significant differentially expressed genes (DEGs) were identified for WRET44.8 mg·mL^−1^, which are known to be involved in tumor suppression and anti-cell proliferation (Table 3).

### 2.6. Pathway Enrichment Analysis 

A KEGG pathway enrichment analysis of the DEGs from each ecotype showed different pathways enriched across the treatments. DEGs with WRE treatment revealed several enriched pathways such as the NF-Kappa B signaling pathway, the TNF signaling pathway, the neuroactive ligand-receptor interaction, Leukocyte transendothelial migration, the IL-17 signaling pathway, the cytokine–cytokine receptor interaction, the metabolic pathway, and the calcium signaling pathway (Appendix A).

## 3. Discussion

### 3.1. Role of Nutraceutical Compounds

Watermelon, notably its rind, is rich in bioactive compounds like lycopene and L-citrulline, which are reported to exhibit antioxidant, anti-diabetic, and anticancer activities [8]. Specifically, WRE has a high concentration of L-citrulline, an amino acid serving as an L-arginine precursor. L-citrulline is an important factor in nitric oxide (NO) synthesis, contributing to various physiological effects, including anticancer activities [7,15]. Our study corroborates these anti-proliferative effects and reveals a 22.29 µg/mg·citrullin concentration in WRE as quantified by LC-MS.

### 3.2. Differential Impact on Apoptosis and Senescence

In the present study, we focused on elucidating the effects of watermelon rind extract (WRE) on various cellular processes, including cell proliferation, apoptosis, senescence, and global transcriptomic alterations in HRAC-769-P cells in vitro. Our findings predominantly indicate that WRE initiates apoptosis rather than senescence in these cells. The poly caspase FLICA probe, FAM-VAD-FMK, utilized in our study, is instrumental in detecting the early stages of apoptosis by identifying activated caspases. Conversely, senescence-associated beta-galactosidase (SA-beta-gal) is an established marker for cell senescence [16]. Our results showed that SA-beta-gal activity remained unchanged or decreased upon treatment with WRE, underscoring a limited role for senescence in this context. These findings confirm that apoptosis and senescence are exclusive cellular outcomes [17].

### 3.3. The Intricacies of Nitric Oxide Signaling

NO is a versatile signaling molecule that induces apoptosis in cancer cells through various mechanisms, including the caspase cascade [18]. Additionally, NO can interact with reactive oxygen species (ROS) to produce peroxynitrite, which has been shown to inflict cellular damage and promote apoptosis [19]. In our study, elevated citrulline levels could be postulated to enhance NO production, thereby triggering apoptosis in HRAC-769-P cells (Figure 6). However, the influence of NO in cellular mechanisms can be complex and context-dependent, necessitating further investigation. Apart from citrulline, alternative mechanisms might involve the inhibition of mitochondrial respiration, changes in cytochrome c levels, or the suppression of anti-apoptotic genes like bcl−2. In summary, while our study establishes a promising link between WRE and induced apoptosis in HRAC-769-P cells, the precise mechanisms and the potential for therapeutic applications remain to be fully explored. The implications of citrulline and NO on different cancer types and cellular contexts also warrant comprehensive investigation.

### 3.4. Watermelon Rind Extract Modulates a Complex Network of Genes

Our study offers valuable insights into the transcriptomic landscape modulated by watermelon rind extract (WRE) in HRAC-769-P cells. The data illuminate how WRE affects various genes that could influence cell proliferation and apoptosis. This complexity underscores the necessity for a systems biology approach to understand the multifaceted gene networks regulated by WRE.

### 3.5. Intrinsic and Extrinsic Apoptotic Pathways

Apoptosis is a well-regulated process involving multiple pathways: the intrinsic (mitochondrial), extrinsic (death receptor), and perforin/granzyme pathways, all of which culminate in caspase-3 activation and cellular degradation [20,21,22]. Our findings, precisely the differential expression of Bmf (Bcl-2-modifying factor) [23] and NFKBIA [24], suggest that WRE predominantly influences the intrinsic apoptotic pathway in HRAC-769-P cells.

### 3.6. Promising Effects of Upregulated Transcripts on Tumor Growth Inhibition

Our transcriptomic data align with existing studies that have identified genes like NPTX1, TRIM31, and CD82/KAI1 as potential therapeutic targets in various cancers [25,26,27,28,29,30,31,32,33]. NPTX1 has been implicated in inhibiting tumor growth across multiple types of cancer, acting through distinct signaling pathways. Similarly, TRIM31 is a tumor suppressor gene in breast and ovarian cancer. ZC3H12A exhibits tumor-suppressive effects in colorectal cancer (CRC) by inducing apoptosis, inhibiting angiogenesis, and EMT signaling. Its expression is associated with chemokine ligands, indicating a potential role of immune response dysregulation in CRC development [34]. Furthermore, MCPIP1, encoded by ZC3h12A-D, hinders cell migration and metastasis through TGF-β signaling inhibition [35]. Linc00472 serves as a tumor suppressor in colorectal and pancreatic cancers. In colorectal cancer, it inhibits proliferation and induces apoptosis by releasing PDCD4 through miR-196a decoying. Linc00472 is a tumor suppressor in colorectal and pancreatic cancers [36,37]. Elevated levels of AATK1, RFN144A, βArr1, and βArr2 KDF1 gene expression lead to various cancer cell inhibitions [38,39,40,41,42,43].

### 3.7. Promising Effects of Downregulated Transcripts on Tumor Growth Inhibition

In addition to the upregulated transcripts, our study also sheds light on the significant downregulation of specific genes reported in various cancer forms. ENC1, SLAMF7 GREB1, EPHA7, CDH6, PKHD1, FBG, FLT3, ADAR2, and ANKRD1 are prominent in this category, suggesting their potential as therapeutic targets in various cancers [44,45,46,47,48,49,50,51,52,53,54,55,56,57,58,59,60,61,62].

### 3.8. Multiple Gene Targets and Potential Therapeutic Avenues

Beyond the well-studied genes, our data indicated the potential roles of several other genes, including PAQR5, TRIM16, and ZC3H12A, in modulating cancer cell behavior [34,35,63,64,65,66]. These genes showed significant changes in expression upon WRE treatment, indicating their role in the anticancer effects of WRE. However, their specific anticancer mechanisms require further investigation.

### 3.9. NF-kappa B and TNF Signaling Pathway

NF-κB and TNF signaling pathways are pivotal in regulating immune responses, inflammation, and cellular processes [67]. Our study suggests that WRE treatment in HRAC-769-P cells may influence these pathways, particularly the NF-κB pathway [68,69,70,71], regulating chemokines, cytokines, and other signaling molecules, potentially leading to anti-proliferative and pro-apoptotic effects.

## 4. Material and Methods

### 4.1. Plant Material and Extraction Process

A fresh fruit of watermelon (*Citrullus lanatus*) was used in the study. The fruit rind or mesocarp was separated and dried. One kg of the dried rind was then finely powdered using a blender, and 100 g of it was dissolved in 70% ethanol in four separate 500 mL aliquots. The derived mixture was filtered using a 0.22 µm bottle top filter (500 mL- CELLTREAT Scientific Products). The filtrate was subjected to evaporation at 50 °C for a few hours to remove the ethanol. The extracted powder was initially dissolved in 0.1% ethanol, and the remaining volume was made up with water, sterilized by filtration, and preserved at −20 °C. The obtained samples were used for various biological and chemical investigations.

### 4.2. Chemical Characterization of Watermelon Rind Extract Using LC-MS

The phytoconstituents of WRE were identified using LC-MS. One mL methanol and 5 µL internal standard (4-Chloro-DL-phenylalanine, 25 μg/mL) were added to the 100 mg·WR sample. This was followed by vortexing for 1 min, centrifugation for 15 min at 20,000 rcf, and 100 µL of liquid fraction was transferred into a glass vial for subsequent analysis. For LC-MS metabolite profiling, a 5 µL sample was injected into the instrument. Samples were analyzed utilizing a Dionex Ultimate 3000 series UHPLC system (Thermo Scientific, Waltham, MA, USA) with a Q-Exactive MS system (Thermo Scientific, Waltham, MA, USA), as described previously [72]. Metabolite assignments were made with citrulline as the target and also as an untargeted LC-MS metabolite profiling assay.

### 4.3. Cell Culture

The HRAC-769-P cells (CRL-1933; American Type Cell Culture, Manassas, VA, USA) were grown in cell culture T75 or T25 flasks (Greiner, Monroe, NC, USA) using 10% FBS (Atlas, Biologicals, Fort Collins, CO, USA) containing RPMI-1640 media and 1% antibacterial-antimycotic solution (Gibco, Grand Island, NY, USA). The cells were kept in a CO_2_ incubator at 37 °C with 90% humidity, using a gas mixture of 5% CO_2_ and 21% O_2_. When the cells reached around 90% confluence, the cells were sub-cultured by splitting them at ratios ranging from 1:4 to 1:12. To achieve this, a 0.25% Trypsin/0.53 mM EDTA solution in Hank’s balanced salt solution (HBSS) without Ca^2+^ and Mg^2+^ (ATCC) was used for 10 min at 37 °C.

### 4.4. Treatments

Different concentrations of stock solutions of WRE were prepared in nuclease and microbial-free water, filter-sterilized, and stored at −20 °C. A stock media for various assays was prepared by adding RPMI-1640 media without phenol red (ATCC) along with FBS (10%), L-glutamine (0.3 g L^−1^; Gibco), and antibiotic-antimycotic solution (1%). The 50% inhibitory concentration (IC50) was considered from dose–response curves using GraphPad Prism 8.4.2 (Appendix A). The IC50 value (88.6 mg·mL^−1^) and another concentration, 44.8 mg·mL^−1^, were combined with the stock media for assays. Corresponding controls were also prepared by mixing similar volumes with the stock media.

### 4.5. Cell Proliferation Assay

Cell proliferation was noted with the WST-8 cell proliferation assay kit (CCK8, Dojindo, Kumamoto, Kyushu, Japan) by using an orange fluorescent dye. WST-8 will be reduced by dehydrogenases abundant in viable cells and transformed to formazan, an orange-colored dye soluble in the culture medium. The amount of formazan dye produced by the activity of dehydrogenases in cells is directly correlated with the number of viable cells. Five hundred live cells were supplemented to each well, holding 100 μL culture media in a 96-well flat black-bottomed plate (Greiner, Monroe, NC, USA). After 24 h of treatment, culture media was substituted with freshly prepared control or treatment media. Plates were collected by gently removing the culture media after 24, 48, and 72 h of treatment and kept at −80 °C before further analysis. Plates were allowed to thaw for 0.5 h at room temperature before starting the assay. In each well, 10 μL of WST-8 (1×) solution was added and thoroughly mixed using a multichannel pipette. The absorbance was then measured at 450 nm (excitation/emission) using a microplate reader (SpectraMax iD3 Multi-Mode Microplate Reader (Molecular Devices, San Jose, CA, USA).

### 4.6. Poly Caspase Assay

The process of apoptosis was performed using the Poly caspase assay kit (FAM FLICA, ImmunoChemistry Technologies, Davis, CA, USA). The probe FAM-VAD-FMK green fluorescent inhibitor binds to active cell caspase enzymes. Cells were grown in T25 flasks with phenol red-free media. Further cells were tested with different treatments when they reached 80% confluence. This study used staurosporine (6 μM) (ImmunoChemistry Technologies, Bloomington, MN, USA.) as a positive control. At various time intervals (0.5, 1, 2, and 4 h), floating cells were collected with media in a 15 mL centrifuge tube at 5000× *g* for 5 min at room temperature. The supernatant was dispensed; cells were suspended with 600 μL of 1× apoptosis wash buffer. Leftover adhered cells on the flask were spooled with trypsin and added to the cell suspension after removing the supernatant. The poly caspase inhibitor FAM-FLICA (1×) was added to the 500 μL cell suspension and allowed for incubation at 37 °C for one hour with intermittent shaking. Simultaneously, some of the remaining cells were utilized to count cell numbers with a hemacytometer using trypan blue (0.04%). Subsequently to the incubation, cells were washed with 2 mL of wash buffer, centrifuged, and the supernatant was eliminated. Following the washing step, the cells were incubated at 37 °C for 12 min to remove any excess FAM-FLICA reagent. Subsequently, the cells were collected through centrifugation, suspended in 500 μL of wash buffer, and kept on ice. Ultimately, 100 μL of cell suspension was utilized for assessing poly-caspase activity in 96-well flat black bottom plates. The optical density (fluorescence) was measured in the microplate reader at 488/520 nm (excitation/emission), and the resulting RFU (Relative Fluorescence Units) values were normalized.

### 4.7. SA-Beta-Gal Assay

Senescence assay was carried out with SA-beta-gal activity with the 96-well cell senescence assay kit (Cell Biolabs, San Diego, CA, USA). The procedure is almost similar to poly-caspase assay except for a few modifications. The activity readings were measured at 0.5, 1, and 2 h. Following the cell collection, they were washed with PBS (phosphate-buffered saline) after the treatment. Subsequently, the cells were suspended in 400 μL of cell lysis buffer (1×) and kept on ice for 10 min. Further, the cell suspension was dissolved thoroughly by vertex, and 100 μL solution was taken out from this and frozen at −80 °C. Moreover, the leftover solution was utilized for the WST-8 cell proliferation assay. A 200 μL solution was mixed with one volume of 2× reaction buffer containing SA-beta-gal substrate and grown at 37 °C for 60 min in the dark. The incubated solution was adequately mixed, and a 200 μL solution was combined with an 800 μL stop solution. A 200 μL solution detected the SA-beta-gal activity in 96-well flat black-bottom plates. The optical density based on fluorescence was measured using a microplate reader at 360/465 nm (excitation/emission), and the resulting RFU (Relative Fluorescence Units) values were normalized or standardized to those obtained from the WST-8 cell proliferation assay.

### 4.8. Wound Healing Assay

The wound healing assay evaluated the effects of cell migration inhibition and metastasis. HRAC-769-P cells were cultured in 6-well plates until reaching 80–90% confluence. Subsequently, uniform scratches/wounds were created in each well using a 20 μL pipette tip. Afterward, the cells were carefully washed with sterile PBS to remove debris and treated with the WRE. The progress of wound closure was observed immediately (0 h) and after 48 h using an inverted microscope (DFC290, Leica, Wetzlar, Germany). Cells were stained with equal volumes of 1% toluidine blue and 1% borax (LabChem Inc., Zelienople, PA, USA) for photography. The experiments were conducted in triplicate to ensure reliability and reproducibility, as described by [73].

### 4.9. RNA Isolation and RNA-Seq Library Preparation

Total ribonucleic acid of control and treatment was extracted from the tissues of biological replicates using RNeasy Plus Mini Kit (QIAGEN, Germantown, MD, USA). The RNA’s quality and quantity were assessed using the bioanalyzer Agilent 2100 and Qubit 4 Fluorometer (Invitrogen, Waltham, MA, USA), respectively. The RNA sequencing libraries were developed using the NEBNext Ultra™ II RNA Library Prep Kit, following the manufacturer’s protocol provided by NEB, USA. The mRNAs were enriched using Oligo (dT) beads, and subsequently, they were fragmented into shorter fragments using fragmentation buffer. The first-strand cDNA was synthesized from the fragmented mRNA using random hexamer primers and later converted into double-strand cDNA. The resulting double-strand cDNAs were end-repaired and added with Illumina sequencing adapters. The adapter-ligated libraries were amplified using sequencing primers for enrichment. The library’s quality and insert size were determined using a bioanalyzer (Invitrogen, Waltham, MA, USA), and the library was estimated using a Qubit fluorometer (Invitrogen, Waltham, MA, USA). The library was diluted to 4 nM concentration and sequenced using Illumina’s NextSeq 500 platform with paired-end sequencing chemistry. The resulting image files in the BCL format were converted to FASTQ with 2 × 150 bp reads using the bcl2fastq tool (Illumina, San Diego, CA, USA).

### 4.10. RNA-Seq Analysis

In the analysis, sequencing adapters and low-quality reads (Phred score QV < 30) were removed using Trimmomatic v. 0.39 [74]. The quality-filtered reads were then mapped to the Human (GRCh38.p13) reference genome (https://www.gencodegenes.org/human/; accessed on 15 April 2023) using STAR RNA-Seq aligner v. 2.7.11a [75] to produce BAM alignment. A read count table was created from the BAM alignment file and genome annotation in GFF format using the HTSeq R package [76]. Differential gene expression (DEG) analysis was carried out using DESeq2 [77], filtering DEGs based on a minimum log2FoldChange of 1 and a false discovery rate (FDR) of 0.05. Pathway Enrichment analyses were conducted using KOBAS (http://bioinfo.org/kobas; accessed on 24 April 2023). 

## 5. Conclusions

Our study reveals that watermelon rind extract (WRE) holds high proportions of nutraceutical components and amino acid derivatives like citrulline and arginine. It demonstrates a dose- and time-dependent reduction in HRAC-769-P cell proliferation in vitro. The decrease in cell proliferation is primarily attributed to apoptosis, supported by the upregulation of early poly-caspase activities and either normal or suppressed SA-beta-gal activity. Transcriptomic analyses suggest that the apoptotic effects may be mediated through both intrinsic and extrinsic pathways involving various key genes and molecular mechanisms. While our study presents a detailed overview, further in-depth research is required to validate these findings and explore clinical applications.

## Figures and Tables

**Figure 1 ijms-24-15615-f001:**
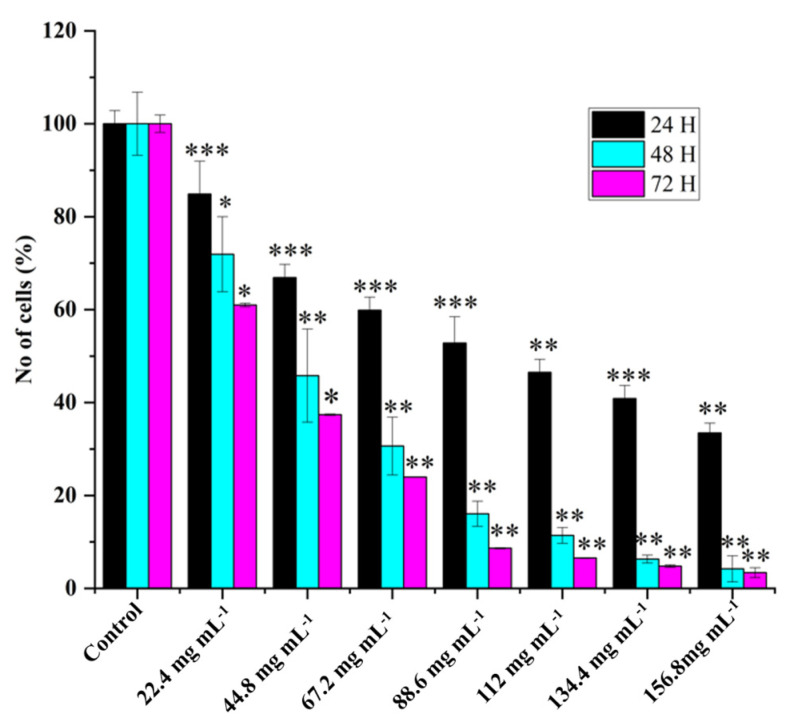
Effect of watermelon rind extract on HRAC-769-P cell proliferation. Percentage of cells at 24, 48, 72 h with various concentrations of WRE (mg·mL^−1^). Control (media without any treatment compound) is adjusted to 100%. Values are means ± SD; *n* = 6; * *p* < 0.05 and ** *p* < 0.01, *** *p* < 0.001 (as compared with control).

**Figure 2 ijms-24-15615-f002:**
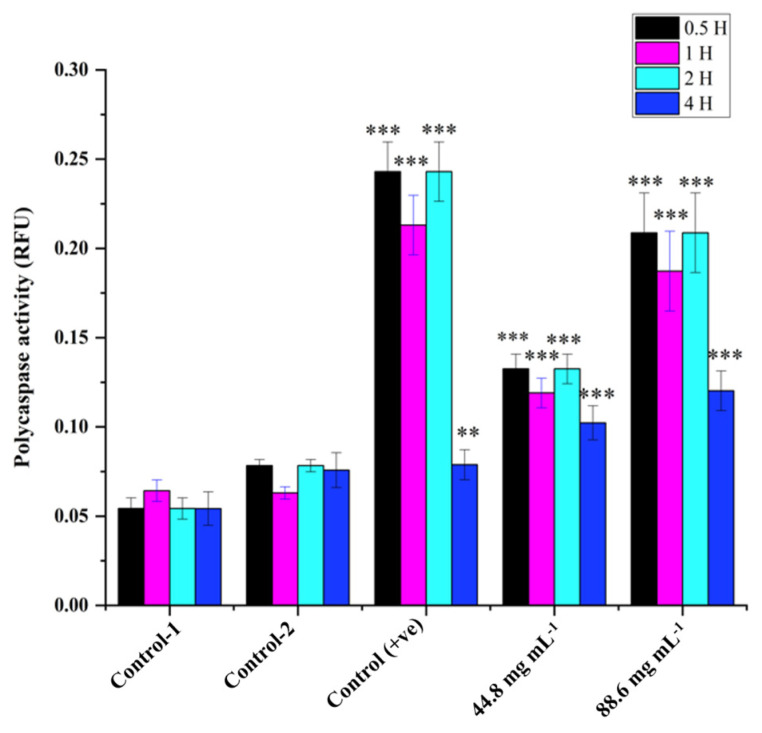
Cellular poly caspase activity in HRAC-769-P cells at 0.5, 1, 2, and 4 h. Control-1 contained culture media. Control-2 contained culture media with 0.96% ethanol. Control (+ve) contained staurosporine (6 μM) 44.8 and 88.6 (mg·mL^−1^) mg of WRE treatment. Values are means ± SD; *n* = 6; ** *p* < 0.01 and *** *p* < 0.001 (as compared with control).

**Figure 3 ijms-24-15615-f003:**
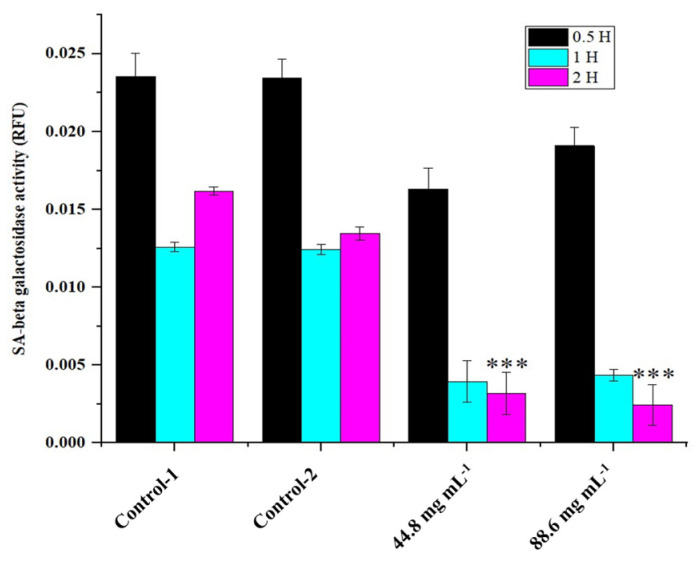
SA-beta-gal activity in HRAC-769-P cells at 0.5, 1, and 2 h. Control-1 contained culture media. Control-2 contained 0.96% ethanol. Control-2 was a corresponding control for WRE 44.8 and 88.6 mg·mL^−1^ treatments. SA-beta-gal activities were normalized with relative fluorescence unit (RFU) values obtained with CyQUANT cell proliferation assay. Values are mean ± SD; *n* = 4; *** *p* < 0.001 (as compared with control).

**Figure 4 ijms-24-15615-f004:**
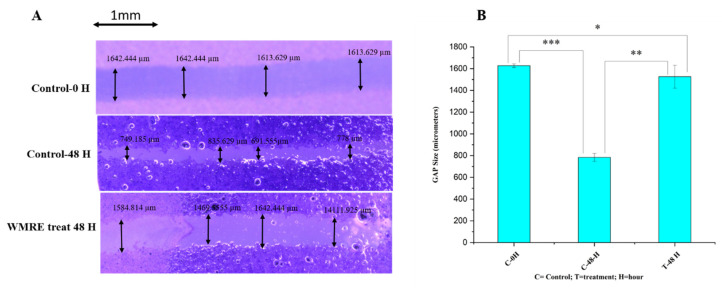
Effect of WRE (88.6 mg·mL^−1^) on cell migration in cells (**A**) Scratching was performed with a 20 µL pipette tip. Quantitative representation of the migration of kidney cancer cells by the wound healing assay. (**B**) The data are presented as the mean and standard deviation. The one-way ANOVA test was used to examine statistical differences. * Significant at *p* < 0.05, ** significant at *p* < 0.001, *** significant at *p* < 0.0001.

**Figure 5 ijms-24-15615-f005:**
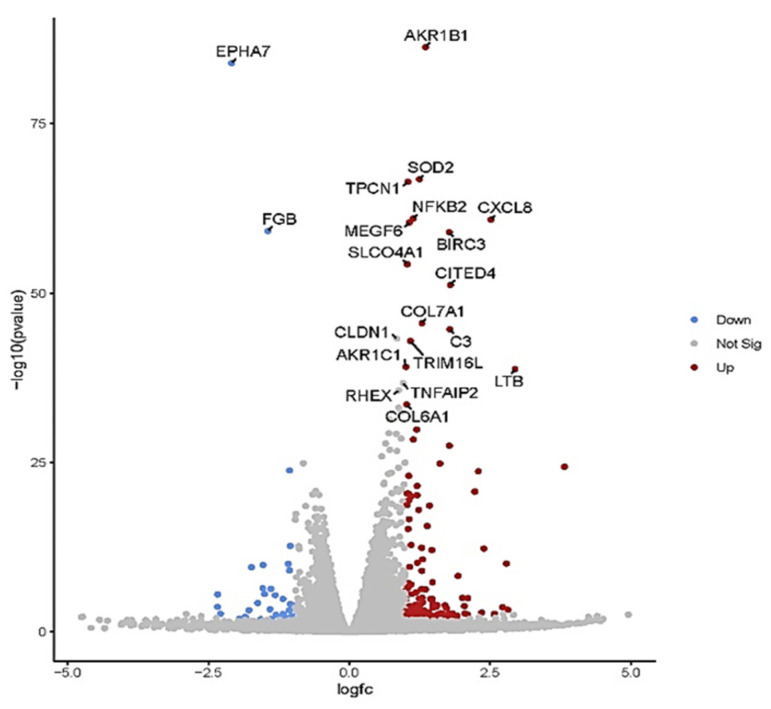
Volcano plot of differentially expressed genes (DEGs). The volcano plots from control vs. treatment conditions 44.8 mg·mL^−1^. The volcano plot illustrates the association between the fold change (log2) and statistical significance (−log10(*p*-value)) of DEGs. The log2 fold change is represented along the *x*-axis, with upregulated genes to the right and downregulated genes to the left. The *y*-axis represents the −log10(*p*-value), with more significant DEGs at the top of the graph. Points on the graph represent individual genes, with color-coding indicating the significance level and fold change: red points represent significantly upregulated genes, blue points represent significantly downregulated genes, and gray points represent non-significant genes.

**Figure 6 ijms-24-15615-f006:**
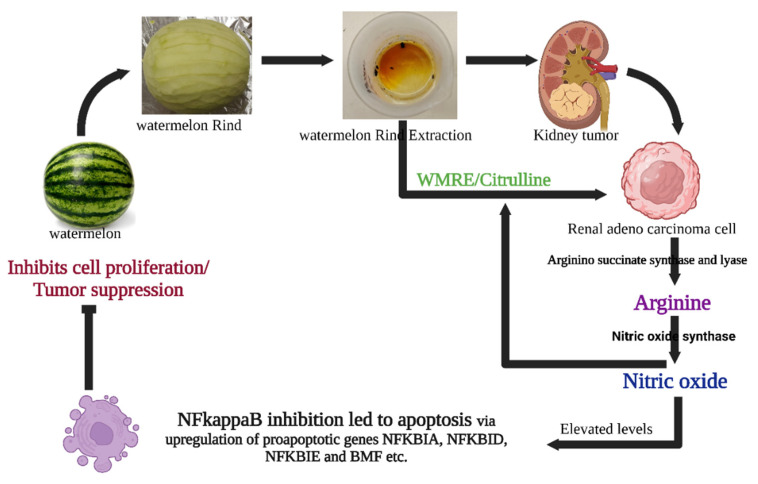
Possible mechanisms of WRE-mediated cell death in HRAC-769-P cells. WRE contains a high amount of citrulline and arginine. L-arginine is produced from L-citrulline to L-arginine by arginine succinate synthease and lyase. Meanwhile, cytokine-induced nitric oxide synthase (iNOS or NOS2) converts arginine into nitric oxide while material is recycled. Nitric oxide can influence survival pathways in cancer cells, leading to a shift in the balance between cell survival and cell death signals in favor of apoptosis. WRE induces apoptosis by regulating the expression of critical genes involved in several cellular pathways, molecular mechanisms, and metabolism. Enzymatic assays and transcriptomic analysis suggested that WRE-induced apoptosis in HRAC-769-P cells was mediated through intrinsic and extrinsic apoptotic pathways and by inhibiting the NF Kappa B pathway and induction of BMF, suppressing anti-apoptotic genes.

**Table 1 ijms-24-15615-t001:** Tentatively identified metabolites via LC-MS from watermelon rind aqueous extract using negative ionization mode. Rt: retention time.

S. No	Proposed Compounds	Formula	Rt [M − H]	Peak Area
	**Amino Acid Derivatives**			
1	4-Methyleneglutamine	C_6_H_10_N_2_O_3_	0.877	2,360,952,958
2	D-(+)-Pyroglutamic Acid	C_5_H_7_NO_3_	1.901	816,414,433.5
3	(2S)-3-(1H-Imidazol-4-yl)	C_12_H_19_N_3_O_7_	1.709	601,125,246
4	DL-Arginine	C_6_H_14_N_4_O_2_	0.85	549,758,768
5	DL-Histidine	C_6_H_9_N_3_O_2_	0.851	432,774,848.5
6	DL-Arginine	C_6_H_14_N_4_O_2_	0.85	549,758,768
7	Ornithine	C_5_H_12_N_2_O_2_	0.882	260,397,275
8	D-(-)-Glutamine	C_5_H_10_N_2_O_3_	9.389	249,168,133.5
9	N-Acetyl-L-Citrulline	C_8_H_15_N_3_O_4_	0.911	13,964,667.5
10	L-(+)-Citrulline	C_6_H_13_N_3_O_3_	3.383	10,651,743
11	Isoleucine	C_6_H_13_NO_2_	2.54	244,248,729
12	L-Phenylalanine	C_9_H_11_NO_2_	3.718	133,133,714
13	L-Glutamic acid, 5-[2-(4-carboxyphenyl)hydrazide]	C_12_H_15_N_3_O_5_	1.71	149,760,416
14	L-(+)-Lactic acid	C_3_H_6_O_3_	10.516	89,115,500
15	D-PANTOTHENIC ACID	C_9_H_17_NO_5_	3.845	143,296,593.5
16	Valine	C_5_H_11_NO_2_	0.996	143,149,592
17	L-Histidine	C_6_H_9_N_3_O_2_	0.837	18,285,235.5
18	(2S)-2-Piperazinecarboxylic acid	C_5_H_10_N_2_O_2_	2.65	29,609,897.5
19	N-Acetylglucosaminitol	C_8_H_17_NO_6_	0.858	23,771,288.5
20	N-Acetyl-L-glutamic acid	C_7_H_11_NO_5_	1.983	22,064,841.5
	Acetylcarnitine	C_9_H_17_NO_4_	8.78	21,886,355.5
21	N-Acetyl-L-phenylalanine	C_11_H_13_NO_3_	5.26	5,322,091
22	**Organic Derivatives**			
23	DL-Malic acid	C_4_H_6_O_5_	0.98	3,630,710,449
24	Isocitric acid	C_6_H_8_O_7_	1.822	1,381,987,018
25	D-(+)-Pyroglutamic Acid	C_5_H_7_NO_3_	1.901	816,414,433.5
26	Anthranilic acid	C_7_H_7_NO_2_	0.912	651,974,104
27	2-(alpha-d-mannosyl)-d-glyceric acid	C_9_H_16_O_9_	9.792	271,529,954.5
28	2-Formyl-1H-pyrrole	C_5_H_5_NO	0.879	261,140,066.5
29	2-(1,3-Benzodioxol-5-yl)-4,5,6,7-tetramethyl-1H-benzimidazole	C_18_H_18_N_2_O_2_	8.301	86,427,528
30	Gluconic acid	C_6_H_12_O_7_	0.881	219,928,281.5
31	2-Furoic acid	C_5_H_4_O_3_	1.433	136,738,393.5
32	Citric acid	C_6_H_8_O_7_	1.822	1,381,987,018
33	Glutarylcarnitine	C_12_H_21_NO_6_	2.963	209,483,451.5
34	Benzyl ë?-primeveroside	C_18_H_26_O_10_	4.177	170,786,527
35	Glucoheptonic Acid	C_7_H_14_O_8_	8.76	43,076,366
36	4-Hydroxybenzoic acid	C_7_H_6_O_3_	4.279	42,979,188
37	Succinic anhydride	C_4_H_4_O_3_	2.332	33,805,020.5
38	Itaconic acid	C_5_H_6_O_4_	1.451	24,903,345.5
39	Mesaconic acid	C_5_H_6_O_4_	1.824	28,480,332
40	Sorbic acid	C_6_H_8_O_2_	0.974	27,447,509.5
41	Malondialdehyde	C_3_H_4_O_2_	0.859	21,556,194.5
42	2,6-Dimethoxybenzoquinone	C_8_H_8_O_4_	0.894	32,512,334
43	Acetonedicarboxylic Acid	C_5_H_6_O_5_		
44	6-Oxo-pipecolinic acid	C_6_H_9_NO_3_	1.702	23,925,387
45	N-Acetyl-L-glutamic acid	C_7_H_11_NO_5_	1.983	22,064,841.5
46	Acetylcarnitine	C_9_H_17_NO_4_	8.78	21,886,355.5
47	Mevalonic acid	C_6_H_12_O_4_	6.95	21,755,237.5
48	Malonic acid			
49	1,3,7-Trimethyluric acid	C_8_H_10_N_4_O_3_	10.067	18,133,176.5
50	5-Hydroxy-2-furoic acid	C_5_H_4_O_4_	1.805	50,812,245.5
51	(±)-Malic Acid	C_4_H_6_O_5_	1.209	50,383,480
52	Glucoheptonic Acid	C_7_H_14_O_8_	8.76	43,076,366
53	Acetylcarnitine	C_9_H_17_NO_4_	8.78	21,886,355.5
54	Itaconic acid	C_5_H_6_O_4_	1.451	24,903,345.5
55	Mesaconic acid	C_5_H_6_O_4_	1.824	28,480,332
56	Sorbic acid	C_6_H_8_O_2_	0.974	27,447,509.5
	Malondialdehyde	C_3_H_4_O_2_	0.859	21,556,194.5
57	Malonic acid			
58	**Sugar Derivatives**			
59	N-Acetylglucosamine	C_17_H_27_N_3_O_17_P_2_	1.264	2,146,760
60	Maltose	C_12_H_22_O_11_	3.179	10,386,453
	Lactose	C_12_H_22_O_11_	0.981	289,280,035.5
61	Sucrose	C_12_H_22_O_11_	10.781	1,531,407
	Trehalose	C_12_H_22_O_11_	0.943	103,537,815
62	Raffinose	C_18_H_32_O_16_	1.32	5,170,093
63	**Hydroxycinnamic Acid Derivatives**			
64	Caffeic acid	C_9_H_8_O_4_	4.098	4,166,656
65	p-Coumaric acid	C_9_H_8_O_3_	5.414	773,133

**Table 2 ijms-24-15615-t002:** Watermelon rind extract-treated HRAC-769-P cells induced transcripts (upregulated and downregulated) mapped to human genome mapping percentage of uniquely mapped reads.

Sample	No. of Raw PE Reads	No. of Filtered PE Reads	No. of Uniquely Mapped PE Reads	Mapping Percentage of Uniquely Mapped Reads
WMRC1	26,784,649	25,630,111	24,702,469	96.4
WMRC2	27,470,824	26,302,571	25,318,621	96.3
WMRC3	25,588,547	24,467,830	23,597,931	96.4
WMRT1	32,424,898	31,024,773	29,939,262	96.5
WMRT2	27,827,568	26,668,858	25,713,431	96.4
WMRT3	31,744,771	30,391,428	29,270,759	96.3
Pairwise comparison		Total DEGs	Upregulated	Downregulated
Control vs. 44.8 MG		186	149	37

**Table 3 ijms-24-15615-t003:** Differentially expressed transcripts of HRAC cells treated with WMR. Fold change of a transcript is a ratio of its expression in control and WMR-treated cells (*n* = 3). The fold change indicates up- (highlighted in blue) or downregulation (highlighted in green) of that transcript, respectively, in WRE-treated cells compared with control.

S. No	Gene ID	Fold Change	*p* adj	Regulation	Annotation
1	NPTX1	3.822151586	1.40205 × 10^−22^	UP	neuronal pentraxin 1(NPTX1)
2	JMA2	3.654043227	6.50348 × 10^−7^	UP	junctional adhesion molecule 2(JAM2)
3	HMOX1	2.791739716	3.72984 × 10^−9^	UP	heme oxygenase 1(HMOX1)
4	TRIM31	2.578310857	0.014206294	UP	tripartite motif containing 31(TRIM31)
5	CXCL2	2.390913463	3.41557 × 10^−11^	UP	C-X-C motif chemokine ligand 2(CXCL2)
6	KDF1	1.864605173	0.049178644	UP	keratinocyte differentiation factor 1(KDF1)
7	TNFAIP3	1.77972605	1.3479 × 10^−25^	UP	TNF alpha induced protein 3(TNFAIP3)
8	EFEMP2	1.753814923	0.045380994	UP	EGF containing fibulin extracellular matrix protein 2(EFEMP2)
9	LINC00887	1.604240482	0.045386292	UP	long intergenic non-protein coding RNA 887(LINC00887)
10	NFKBID	1.4789834	1.08647 × 10^−6^	UP	NFKB inhibitor delta(NFKBID)
11	LINC00472	1.467452434	0.000148121	UP	long intergenic non-protein coding RNA 472(LINC00472)
12	PAQR5	1.355607471	9.19446 × 10^−6^	UP	progestin and adipoQ receptor family member 5(PAQR5)
13	COL7A1	1.292376567	2.89259 × 10^−43^	UP	collagen type VII alpha 1 chain(COL7A1)
14	RNF144B	1.287668834	8.06605 × 10^−6^	UP	ring finger protein 144B(RNF144B)
15	SOD2	1.24731498	6.67748 × 10^−64^	UP	superoxide dismutase 2(SOD2)
16	ARRB1	1.244050952	0.00013318	UP	arrestin beta 1(ARRB1)
17	BMF	1.183513353	0.000755165	UP	Bcl2 modifying factor(BMF)
18	NFKBIA	1.140983535	1.77033 × 10^−26^	UP	NFKB inhibitor alpha(NFKBIA)
19	TPD52L1	1.162282589	0.007265863	UP	TPD52 like 1(TPD52L1)
20	RHBDL1	1.127426241	0.039940739	UP	rhomboid like 1(RHBDL1)
21	CD82	1.109222868	3.56575 × 10^−5^	UP	CD82 molecule(CD82)
22	CLIP4	1.105718912	0.004748612	UP	CAP-Gly domain containing linker protein family member 4(CLIP4)
23	TRIM16L	1.088105329	9.69946 × 10^−41^	UP	tripartite motif containing 16 like(TRIM16L)
24	NFKBIE	1.065145553	3.25882 × 10^−15^	UP	NFKB inhibitor epsilon(NFKBIE)
25	TMEM158	1.061667088	1.49615 × 10^−6^	UP	transmembrane protein 158(TMEM158)
26	ZC3H12A	1.058928366	7.96899 × 10^−18^	UP	s100 calcium binding protein A4(S100A4)
27	HSF4	1.058853794	0.005924004	UP	heat shock transcription factor 4(HSF4)
28	AKR1C2	1.058573734	2.63416 × 10^−21^	UP	aldo-keto reductase family 1 member C2(AKR1C2)
29	ADAMTS7	1.047212012	7.15087 ×10^−14^	UP	ADAM metallopeptidase with thrombospondin type 1 motif 7(ADAMTS7)
30	KCNK3	1.026865059	3.68494 × 10^−17^	UP	potassium two pore domain channel subfamily K member 3(KCNK3)
31	PGGHG	1.026377088	9.09732 × 10^−19^	UP	protein-glucosylgalactosylhydroxylysine glucosidase(PGGHG)
32	LACTB	1.007908666	0.004812089	UP	lactamase beta(LACTB)
33	ATG16L2	1.001999864	1.37778 × 10^−5^	UP	autophagy related 16 like 2(ATG16L2)
34	CELF2	−1.012165448	0.00091243	DOWN	CUGBP Elav-like family member 2(CELF2)
35	KCNH1	−1.01380309	0.020077479	DOWN	potassium voltage-gated channel subfamily H member 1(KCNH1)
36	CDH6	−1.044599459	1.57671 × 10^−11^	DOWN	cadherin 6(CDH6)
37	DOCK2	−1.058742034	4.76266 × 10^−22^	DOWN	dedicator of cytokinesis 2(DOCK2)
38	PAX8-AS1	−1.058885169	2.95114 × 10^−8^	DOWN	PAX8 antisense RNA 1(PAX8-AS1)
39	ANKRD1	−1.080370398	4.09198 × 10^−9^	DOWN	ankyrin repeat domain 1(ANKRD1)
40	TAF1A-AS1	−1.086638886	0.038122062	DOWN	TAF1A antisense RNA 1(TAF1A-AS1)
41	RHOU	−1.10777548	0.048963406	DOWN	ras homolog family member U(RHOU)
42	SLC16A9	−1.1715508	0.010549451	DOWN	solute carrier family 16 member 9(SLC16A9)
43	ENC1	−1.17558341	0.000148121	DOWN	ectodermal-neural cortex 1(ENC1)
44	GREB1	−1.200291699	0.038699026	DOWN	growth regulating estrogen receptor binding 1(GREB1)
45	PKHD1	−1.311859039	4.96206 × 10^−5^	DOWN	PKHD1 ciliary IPT domain containing fibrocystin/polyductin(PKHD1)
46	HORMAD2-AS1	−1.375725708	0.042051375	DOWN	HORMAD2 and MTMR3 antisense RNA 1(HORMAD2-AS1)
47	FGB	−1.441683784	1.07928 ×10^−56^	DOWN	fibrinogen beta chain(FGB)
48	SLCO4C1	−1.500449851	3.46892 × 10^−5^	DOWN	solute carrier organic anion transporter family member 4C1(SLCO4C1)
49	NPNT	−1.529088042	5.81654 × 10^−9^	DOWN	nephronectin(NPNT)
50	ARHGAP28	−1.53301957	6.57424 × 10^−6^	DOWN	Rho GTPase activating protein 28(ARHGAP28)
51	C1orf116	−1.56932167	0.045320324	DOWN	chromosome 1 open reading frame 116(C1orf116)
52	SLAMF7	−1.840054142	0.024930087	DOWN	SLAM family member 7(SLAMF7)
53	UNC13C	−1.944259269	0.03671916	DOWN	unc-13 homolog C(UNC13C)
54	EPHA7	−2.087931878	7.25051 × 10^−81^	DOWN	EPH receptor A7(EPHA7)
55	SLC26A5-AS1	−2.285084696	0.010311261	DOWN	SLC26A5 antisense RNA 1(SLC26A5-AS1)
56	SULT1B1	−2.333807748	3.9407 × 10^−5^	DOWN	sulfotransferase family 1B member 1(SULT1B1)
57	FLT3	−2.335935596	0.001480949	DOWN	fms related receptor tyrosine kinase 3(FLT3)

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
