# Peer review of "From Fruit Waste to Medical Insight: The Comprehensive Role of Watermelon Rind Extract on Renal Adenocarcinoma Cellular and Transcriptomic Dynamics"

_ijms, 2023, doi:10.3390/ijms242115615_

Round 1

Reviewer 1 Report

The manuscript by Reddy et al reported the effects of watermelon rind extract on cell proliferation, apoptosis, senescence, and transcriptome profiles of human renal adenocarcinoma cells. WR metabolome analysis identified targeted citrulline (22.29 µg/mg) and untargeted other metabolites in our study, including amino acid derivatives, organic acid derivatives, sugar derivatives, and hydroxycinnamic acid derivatives. In vitro cell experiments indicated various concentrations of WRE extract's effect on HRAC-769-P cell proliferation showed dose-dependent cell viability. The decrease in cell proliferation is primarily attributed to apoptosis, supported by upregulation of early poly-caspase activities and either normal or suppressed SA-beta-gal activity. Transcriptomic analyses suggest that the apoptotic effects may be mediated through both intrinsic and extrinsic pathways involving various key genes and molecular mechanisms. This present study is novelty, but the overall experimental design is not rigorous enough. For example, the mechanism of action of watermelon rind extract in inhibiting the proliferation of renal cancer cells has not been verified experimentally. Furthermore, expression of related proteins was not confirmed by Western blotting experiments. To sum up, this manuscript is not suitable for publication in International Journal of Mechanical Sciences.

No

Author Response

We appreciate the detailed feedback provided by Reviewer 1 regarding our manuscript. Our study presented a holistic approach by utilizing multi-omics data, encompassing both metabolomics and transcriptomics, to understand the impact of watermelon rind extract (WRE) on renal cancer cells. We meticulously examined cell viability differentials between experimental units (control vs. treatment). We also utilized a poly-caspase assay to ascertain the mechanism of apoptosis and an SA-beta-gal assay to probe senescence. The outcomes of these assays, which emulate protein assays, offered a clear insight into the dynamics of programmed cell death, leading us to conclude that apoptosis plays a central role in the observed inhibition of cancer cell proliferation.Moreover, our study presented compelling evidence supporting the inhibitory effects of WRE on metastasis, demonstrated through wound healing assays. While we concur with the reviewer that the explicit mechanism of action of WRE was not delineated in exhaustive detail, our primary objective was to gauge the overarching effects of WRE as a whole rather than focusing on individual metabolites. However, acknowledging the valid concerns raised, we are committed to further refining our experimental approach in future studies. We plan to isolate and test individual compounds from WRE, and we will incorporate additional experimental methodologies, such as Western blotting, to corroborate protein expression. We are sincerely grateful to Reviewer  for the constructive criticism and recommendations, which will undoubtedly aid in enhancing the robustness and depth of our study.

Reviewer 2 Report

The work here presented has been performed according to a rigorous experimental design applying several genetic and molecular biology methodologies. The obtained results are interesting and the conclusions drawn are well supported. The manuscript is suitable for being accepted in the journal, provided that the following discrepancy is addressed.

When the authors talk about the activity of SA-beta-gal they seem to argue that the treatment is similar to the control at 0.5 and 1 h, with a decrease in activity starting from 2 hours (page 4, paragraph 2.3, lines 106-109). Instead, looking at Figure 3, treated and control show a statistical significant difference already at 0.5 h. They resume this statement at page 9 (paragraph 3.2, lines 201-202) of the Discussion section. Another possibility is that the authors mistakenly placed asterisks indicating statistical significance above the black bars (0.5 hours). In this case, by correcting figure 3, the text is correct.

Author Response

We appreciate the reviewer's feedback and constructive review, and we have made the necessary changes per the suggestion.

Reviewer 3 Report

In this manuscript, "Unveiling the Potential of Watermelon Rind Extract: Impact on Cell Proliferation, Apoptosis, Senescence, and Transcriptomic Profile in Human Renal Adenocarcinoma Cells," the author explored antitumor properties of watermelon rind extract on tumor cells of renal adenocarcinoma. I have flowing queries/suggestions.

1.     The manuscript lacks a precise mechanism of action, and key molecules of watermelon rind implicate anticancer activity. It mainly reports the phenomenon following tumor cell treatment with WRE rather than specific molecular mechanisms.

2.     The author listed many micromolecules in watermelon rind through LC/MS study. However, which one (more) is responsible for antitumor activity is missing? The author can include the results of purified citrulline's effect on these cancer cells if they suggest citrulline is the main active compound in WRE.

3.     The molecular mechanism explained in this study is primarily based on RNA-seq data, which needs further validation through molecular techniques, e.g., qPCR. Author can include RNA/protein expression of crucial stakeholders, e.g., Bmf, NFKBIA, NPTX1, TRIM31, CD82/KAI1, ENC1, SLAMF7 GREB1, EPHA7, CDH6, PKHD1, FBG, FLT3, ADAR2, ANKRD1 and mediator NF-kappa B and TNF signaling pathway through qPCR or western blotting or ELISA.

4.     The study correlates high citrulline increases NO levels. Rather than connecting indirectly, the author can check NO levels in treated and untreated cells.

Author Response

  1. The manuscript lacks a precise mechanism of action, and key molecules of watermelon rind implicate anticancer activity. It mainly reports the phenomenon following tumor cell treatment with WRE rather than specific molecular mechanisms.

A: We acknowledge the reviewer's point emphasizing the importance of elucidating specific molecular mechanisms behind the observed effects of watermelon rind extract (WRE) on tumor cells. In our current research, we have adopted a holistic approach by integrating a wide spectrum of multi-omics data. This study encompasses both metabolomics and transcriptomics, which, when coupled with physiological data, provides an overarching view of the response in renal cancer cells when exposed to WRE. While this approach offers a broader understanding of the cellular response, we recognize the need for a more granular exploration of the specific molecular players and pathways. Moving forward, we are committed to deepening our investigation. Our future research endeavors will focus on isolating and studying the effects of individual purified metabolites from WRE. This research will help us pinpoint the specific molecules and pathways instrumental in mediating the anticancer effects. We appreciate the reviewer's insightful feedback, which will be instrumental in guiding the direction of our subsequent research in this domain.

  1. The author listed many micromolecules in watermelon rind through LC/MS study. However, which one (more) is responsible for antitumor activity is missing? The author can include the results of purified citrulline's effect on these cancer cells if they suggest citrulline is the main active compound in WRE.

A. Thank you for the insightful observation. We concur that understanding the specific role of individual molecules in the antitumor activity is crucial for a deeper comprehension of the therapeutic potential of WRE. It's worth noting that our focus on citrulline was primarily inspired by prior research. Specifically, a study conducted by El Gizawy et al. in 2022 provided compelling evidence that citrulline could inhibit the proliferation of several distinct cancer cell lines. Drawing from this evidence, we gave particular attention to citrulline in our analysis. However, as pointed out by the reviewer, we recognize that the watermelon rind contains many potential antitumor compounds. The observed effects in our assays might arise from the synergistic or additive interactions of these various compounds rather than one singular molecule. To further elucidate the contributions of individual compounds, we have plans to embark on a focused investigation. Our future endeavors will involve isolating and testing the various anticancer compounds detected in our current study. This approach will allow us to more accurately identify and quantify the therapeutic potential of each compound, potentially leading to a combination of key molecules that exert the most potent antitumor effects.We are grateful for the constructive feedback, as it emphasizes the importance of delineating the specific roles of individual molecules in complex extracts like WRE.

  1. The molecular mechanism explained in this study is primarily based on RNA-seq data, which needs further validation through molecular techniques, e.g., qPCR. Author can include RNA/protein expression of crucial stakeholders, e.g., Bmf, NFKBIA, NPTX1, TRIM31, CD82/KAI1, ENC1, SLAMF7 GREB1, EPHA7, CDH6, PKHD1, FBG, FLT3, ADAR2, ANKRD1 and mediator NF-kappa B and TNF signaling pathway through qPCR or western blotting or ELISA.

Thank you for your astute observation and suggestions. We recognize the importance of cross-validating and substantiating RNA-Seq findings through complementary molecular techniques to ensure robustness and authenticity in the data. Our current RNA-Seq experiment was meticulously conducted with three distinct biological replicates. This rigorous approach gave us statistically significant insights into the differentially expressed genes (DEGs) in response to WRE treatment. While the primary intent of this study was to delineate the WRE-specific metabolome and the overarching transcriptome changes in cancer cells, we acknowledge the merit of drilling down further to validate the expression of key molecular entities. In alignment with your recommendations, we have charted a future research path. Our forthcoming investigations will focus on isolating and assessing compounds showcasing potential antitumor activities. We are also committed to incorporating molecular validation techniques such as Western blotting or ELISA assays to validate the expression of the key genes and proteins highlighted. Your feedback has been instrumental in refining our research trajectory, and we sincerely appreciate your insights and expertise.

  1. The study correlates high citrulline increases NO levels. Rather than connecting indirectly, the author can check NO levels in treated and untreated cells.

Thank you for your constructive feedback. We concur with your perspective that direct measurement of NO levels in treated versus untreated cells would offer more definitive and tangible evidence. Our current research primarily delved into the downstream regulation of citrulline, extrapolating potential pathways based on KEGG analysis and previously published research. While our focus was on global transcriptome analysis, we understand the inherent challenges in attributing specific effects to a single compound amidst the complexities of cellular responses. Your input underscores experimental validation's importance in corroborating our hypotheses and strengthening the study's conclusions. As we continue our research in this area, we will consider direct assessments, such as measuring NO levels, to provide a more comprehensive understanding of the mechanisms at play. We deeply appreciate your thorough review and the insights you've provided, which will undoubtedly guide our future research endeavors.

Round 2

Reviewer 3 Report

 None

Author Response

Thank you for your detailed review and insightful comments on our manuscript. We value your feedback, which has provided clarity to areas that required revision.

Regarding the discrepancy between the written description and Figure 3 for the activity of SA-beta-gal, we have thoroughly re-examined our data. Upon verification, it was identified that there was an error in the placement of asterisks in Figure 3, which indicated statistical significance at 0.5 hours. We have rectified this error, and the corrected Figure 3 now aligns with the description provided in the text. We sincerely apologize for any confusion that this oversight might have caused. We are grateful for your sharp observation which has enabled us to make the necessary corrections and improve the quality of our manuscript. The rectified Figure 3 and the accompanying textual references have been updated in the revised manuscript for clarity and consistency.

We hope that with these changes, our manuscript meets the journal's standards for acceptance. Once again, thank you for your valuable input and for helping us improve our work.